# Histone Demethylase KDM7A Contributes to the Development of Hepatic Steatosis by Targeting Diacylglycerol Acyltransferase 2

**DOI:** 10.3390/ijms222011085

**Published:** 2021-10-14

**Authors:** Ji-Hyun Kim, Arukumar Nagappan, Dae Young Jung, Nanjoo Suh, Myeong Ho Jung

**Affiliations:** 1Division of Longevity and Biofunctional Medicine, School of Korean Medicine, Pusan National University, 49 Busandaehak-ro, Mulgeum-eup, Yangsan-si, Gyeongnam 50612, Korea; kimji77@pusan.ac.kr (J.-H.K.); arulbiotechtnau@gmail.com (A.N.); dyjung999@naver.com (D.Y.J.); 2Department of Chemical Biology, Ernest Mario School of Pharmacy, Rutgers, The State University of New Jersey, 164 Frelinghuysen Road, Piscataway, NJ 08854, USA; nsuh@pharmacy.rutgers.edu

**Keywords:** diacylglycerol acyltransferase 2, histone demethylase KDM7A, histone methylation, hepatic steatosis, nonalcoholic fatty liver disease, triglyceride

## Abstract

Nonalcoholic fatty liver disease (NAFLD) is the most common chronic liver disease. While the development of NAFLD is correlated with aberrant histone methylation, modifiers of histone methylation involved in this event remain poorly understood. Here, we studied the functional role of the histone demethylase KDM7A in the development of hepatic steatosis. KDM7A overexpression in AML12 cells upregulated diacylglycerol acyltransferase 2 (DGAT2) expression and resulted in increased intracellular triglyceride (TG) accumulation. Conversely, KDM7A knockdown reduced DGAT2 expression and TG accumulation, and significantly reversed free fatty acids-induced TG accumulation. Additionally, adenovirus-mediated overexpression of KDM7A in mice resulted in hepatic steatosis, which was accompanied by increased expression of hepatic DGAT2. Furthermore, KDM7A overexpression decreased the enrichment of di-methylation of histone H3 lysine 9 (H3K9me2) and H3 lysine 27 (H3K27me2) on the promoter of DGAT2. Taken together, these results indicate that KDM7A overexpression induces hepatic steatosis through upregulation of DGAT2 by erasing H3K9me2 and H3K27me2 on the promoter.

## 1. Introduction

Nonalcoholic fatty liver disease (NAFLD) is recognized as the most common chronic liver disease with an increasing prevalence, and is strongly associated with obesity, insulin resistance, cardiovascular disease, hypertriglyceridemia, and hyperuricemia [1]. NAFLD encompasses a broad spectrum ranging from mild hepatic steatosis to more severe forms such as nonalcoholic hepatitis (NASH), hepatic cirrhosis, and hepatocarcinoma [1]. Hepatic steatosis, defined as triglyceride (TG) accumulation in the cytoplasm of hepatocytes, is a driving force for NAFLD [1]. Hepatic steatosis arises from the imbalance between TG acquisition involving de novo lipogenesis (DNL) and uptake of fatty acid (FA) from the adipose tissue, and TG removal involving β-oxidation of FA and export of very low-density lipoprotein from the liver [1]. NASH develops from hepatic steatosis by a combination of TG accumulation, apoptosis, oxidative stress, inflammation, and fibrosis, leading to hepatic cirrhosis, hepatocarcinoma, and ultimately liver-related mortality [1].

While various factors, including genetic disposition, contribute to the development of NAFLD, alteration of the epigenome by environmental factors can play a key role in this event. Several studies have reported the link between the regulation of hepatic steatosis and histone modifications [2,3,4]. Histone deacetylase 3 (HDAC3) is an epigenetic regulator associated with hepatic steatosis [2]. HDAC3 downregulates the expression of genes related to fatty acid uptake, TG synthesis, and lipolysis, thus preventing hepatic steatosis [2]. Furthermore, the histone H3K4 methyltransferase MLL3/4 upregulates proliferator-activated receptor gamma isoform 2 (PPARγ2) and its target steatosis genes, thereby inducing hepatic steatosis [3]. The histone demethylase plant homeodomain finger 2 (Phf2) was recently shown to induce hepatic steatosis through the upregulation of carbohydrate-responsive element-binding protein (ChREBP) target genes [4]. Our previous study demonstrated that the histone H3K9 demethylase JMJD2B stimulated the expression of PPARγ2 and its target genes and contributed to hepatic steatosis [5]. However, the epigenetic regulation of hepatic steatosis through histone methylation is still poorly understood.

The histone lysine demethylase KDM7A (also known as JHDM1D) is a plant homeodomain (PHD) finger protein (PHF) and Jumonji (Jmj)-containing histone demethylase [6]. KDM7A is a dual histone demethylase that removes repressive di-methylation marks at lysine 9 of histone H3 (H3K9me2) and lysine 27 of histone H3 (H3K27me2), converting the marks to a mono-methylated state; thus, KDM7A functions as a transcription activator [6]. Reportedly, KDM7A promotes neural differentiation through the upregulation of CCAAT-enhancer-binding protein β and fibroblast growth factor 4 expression [7], and balanced adipogenic and osteogenic differentiation from progenitor cells by controlling Wnt signaling [8]. Furthermore, KDM7A is involved in promoting prostate cancer [9]. KDM7A was recently shown to control inflammatory responses through the regulation of nuclear factor kappa B (NF-κB)-dependent genes [10]. However, the role of KDM7A in the development of hepatic steatosis has not been elucidated.

Two diacylglycerol acyltransferases (DGATs), namely, DGAT1 and DGAT2, are known to play an important role in hepatic TG synthesis and induce hepatic steatosis through the catalysis of the final step of TG synthesis [11]. In the liver, these two enzymes differentially use exogenously or endogenously synthesized fatty acids. While DGAT1 preferentially utilizes exogenously fatty acids for TG synthesis, DGAT2 consumes fatty acids from DNL and increases in NAFLD. Using RNA-sequencing study, we found that the expression of DGAT2 was increased in KDM7A-overexpressing AML12 cells. Accordingly, we hypothesized that KDM7A epigenetically upregulates DGAT2 expression by erasing the transcriptionally repressive histone marks H3K9me2 and H3K27me2 on the promoter of DGAT2, eventually leading to hepatic steatosis through increasing TG synthesis.

In the present study, we investigated whether KDM7A plays a role in the development of hepatic steatosis through the upregulation of DGAT2 in vitro and in vivo. We measured intracellular TG accumulation and DGAT2 expression in KDM7A-overexpressing or knockdown AML12 cells. In addition, we examined the enrichment of repressive histone marks H3K9me2 and H3K27me2 on the promoter of DGAT2 in KDM7A-overexpressing AML12 cells and assessed the development of hepatic steatosis in vivo in KDM7A-overexpressing mice.

## 2. Results

### 2.1. Expression of KDM7A and DGAT2 Is Increased in Hepatosteatosis Cell and Animal Models

To determine whether the expression of KDM7A and DGAT2 correlates with hepatic steatosis pathogenesis, we examined the expression of KDM7A and DGAT2 in hepatic steosis-induced cell and animal models. To produce a hepatic steatosis cell model, HepG2 and AML12 cells were treated with a mixture of free fatty acids (FFAs) (oleic acid: palmitic acid, 2:1). FFAs treatment increased the intracellular levels of TG in both cells as compared to control (FFAs-untreated cells) treatment (Figure 1A,B). This observation was consistent with the significant increase in the protein and mRNA levels of KDM7A and DGAT2 in FFAs-treated HepG2 and AML12 cells (Figure 1A,B). Furthermore, we examined the expression of KDM7A and DGAT2 in the liver tissue of high-fat diet (HFD)-fed obese mice. Concomitant with increased TG accumulation (Figure 1C), Western blot analysis revealed that protein levels of KDM7A and DGAT2 increased in the liver of HFD-fed obese mice more than in the liver of lean mice (Figure 1C). qPCR analysis also confirmed increased mRNA levels of KDM7A and DGAT2 in the liver of HFD-fed obese mice (Figure 1C). These results suggest that the increased expression of KDM7A and DGAT2 may contribute to the development of hepatic steatosis.

### 2.2. KDM7A Overexpression Upregulates DGAT2 and Increases Intracellular TG Levels in AML12 Cells

To determine whether KDM7A induces hepatic steatosis through the upregulation of DGAT2 expression, a gain-of-function study was performed with AML12 cells. AML12 cells were infected with an adenovirus carrying KDM7A (Ad-KDM7A) for ectopic overexpression of KDM7A, and intracellular TG levels were measured. To verify KDM7A overexpression in Ad-KDM7A-infected AML12 cells, the protein and mRNA levels of KDM7A were analyzed. Western blot and qPCR analysis revealed that KDM7A levels were significantly higher in Ad-KDM7A-infected AML12 cells than in Ad-GFP-infected AML12 cells (Figure 2A,B). KDM7A is a dual histone demethylase for H3K9me2 and H3K27me2 [7]. Thus, we determined the levels of global H3K9me2 and H3K27me2 in Ad-KDM7A-infected AML12 cells. Consistent with the increased KDM7A expression, Ad-mediated KDM7A overexpression led to a decrease in H3K9me2 and H3K27me2 global epigenetic marks, demonstrating the successful overexpression of KDM7A in Ad-KDM7A-infected AML12 cells (Figure 2A). Then, we examined the protein and mRNA levels of DGAT2 in Ad-KDM7A-infected AML12 cells to see whether KDM7A regulates DGAT2 expression. Western blot and qPCR analysis revealed a significant increase in the protein and mRNA levels of DGAT2 by KDM7A overexpression (Figure 2A,B). We next measured intracellular TG levels in Ad-KDM7A-infected AML12 cells. Quantitation of TG levels showed that KDM7A overexpression increased intracellular TG levels in AML12 cells (Figure 2C). Then, to investigate whether the KDM7A-induced TG levels is mediated via DGAT2, TG levels were assessed in KDM7A-overexpressing AML12 cells treated with PF-06427878, DGAT2 inhibitor. Although PF-06427878 itself reduced basal TG levels in control AML12 cells infected with Ad-GFP, PF-06427878 treatment to Ad-KDM7-infected AML12 cells significantly blocked KDM7A-induced TG levels (Figure 2C). Taken together, these results suggest that KDM7A contributes to the induction of TG accumulation through the upregulation of DGAT2.

### 2.3. KDM7A Knockdown Downregulates DGAT2 and Decreases Intracellular TG Levels in AML12 Cells

To further confirm the functional role of KDM7A in hepatic steatosis, a loss-of-function study was performed with AML12 cells. AML12 cells were transfected with an adenovirus carrying shRNA against KDM7A (Ad-KDM7A shRNA). To verify the knockdown of KDM7A, the expression of KDM7A and its target histone marks were measured in Ad-KDM7A shRNA-infected AML12 cells. Western blot and qPCR analysis revealed that the KDM7A shRNA efficiently reduced the KDM7A level (Figure 3A,B). Consistent with the reduced KDM7A level, KDM7A knockdown increased the H3K9me2 and H3K27me2 global epigenetic marks (Figure 3A), indicating that KDM7A expression was successfully knocked down in KDM7A shRNA-infected AML12 cells. We then assessed the expression of DGAT2 in KDM7A shRNA-infected AML12 cells and found that KDM7A knockdown decreased the protein and mRNA levels of DGAT2 (Figure 3A,B). We also measured intracellular TG levels in KDM7A shRNA-infected AML12 cells. As shown in Figure 3C, KDM7A knockdown significantly reduced intracellular TG levels in AML12 cells. Furthermore, we investigated the effects of KDM7A knockdown on FFAs-induced TG accumulation. As shown in Figure 3D, FFAs treatment increased the intracellular accumulation of TG; however, this increase was blocked after KDM7A knockdown. Taken together, these results indicate that KDM7A upregulates the expression of DGAT2, thereby possibly playing a key role in the development of hepatic steatosis.

### 2.4. KDM7A Reduces the Enrichment of H3K9me2 and H3K27me2 on the Promoter of DGAT2

Next, we explored the potential mechanism underlying KDM7A-mediated upregulation of DGAT2 expression. As KDM7A is a dual histone H3K9me2 and H3K27m2 demethylase, we hypothesized that KDM7A demethylates H3K9me2 and H3K27me2 on the promoter of DGAT2, leading to the stimulation of DGAT2 expression. To prove this, we examined the recruitment of KDM7A and enrichment of H3K9me2 and H3K27me2 onto the promoter of DGAT2 in KDM7A-overexpressing AML12 cells. As shown in Figure 4A, ChIP-qPCR assay revealed a significant increase in KDM7A recruitment onto DGAT2 promoter in KDM7A-overexpressing cells compared with that in control Ad-GFP-infected AML12 cells (Figure 4A). Concomitant with increased recruitment of KDM7A, the enrichment of H3K9me2 and H3K27me2 on the promoter of DGAT2 was reduced in Ad-KDM7A-infected AML12 cells (Figure 4B), indicating that KDM7A removes the repressive histone marks, H3K9me2 and H3K27me2, on the promoter of DGAT2 and stimulates DGAT2 expression.

### 2.5. Adenovirus-Mediated KDM7A Overexpression Induces Hepatic Steatosis In Vivo

To understand the functional role of KDM7A in hepatic steatosis in vivo, we injected recombinant Ad-KDM7A through the tail vein of C57BL/6J mice to overexpress hepatic KDM7A. The mice were then fed a ND for 4 weeks. To investigate the effect of KDM7A on the development of hepatic steatosis, liver TG level and lipid droplets were examined in Ad-KDM7A-injected mice. Quantitation of hepatic TG revealed a higher TG level in Ad-KDM7A-injected mice than in Ad-GFP-injected mice (Figure 5A), consistent with the in vitro data that KDM7A overexpression increased the intracellular TG level in AML12 cells. The appearance of the liver revealed that Ad-KDM7A-injected mice increased steatosis (Figure 5B). H&E staining showed that Ad-KDM7A-injected mice had more lipid droplets than Ad-GFP-injected mice (Figure 5C). ORO staining also confirmed the marked increase in hepatic TG level in Ad-KDM7A-injected mice compared with that in Ad-GFP-injected mice (Figure 5C). Consistent with the observed phenotypes, the biochemical analysis demonstrated that the serum levels of TG, free fatty acids, and total cholesterol were higher in Ad-KDM7A-injected mice than in Ad-GFP-injected mice (Figure 5D). We confirmed whether KDM7A was successfully overexpressed in Ad-KDM7A-injected mice. Western blot and qPCR analysis revealed that the protein and mRNA levels of KDM7A increased in the liver of Ad-KDM7A-injected mice (Figure 5E). Consistent with the in vitro data, KDM7A overexpression also increased the protein and mRNA levels of DGAT2 in the liver of Ad-KDM7A-injected mice (Figure 5E). Taken together, our results indicate that KDM7A overexpression in the liver induces hepatic steatosis in mice.

## 3. Discussion

Epigenetic modifiers can contribute to the development and progression of several metabolic diseases, including NAFLD [12]. Understanding of the epigenetic mechanisms involved in NAFLD progression may reveal therapeutic targets; therefore, identifying epigenetic modifiers is imperative to make progress in the treatment of NAFLD. Aberrations in histone methylation are known to correlate with the development of hepatic steatosis [2,3,4,5]. However, histone methylation underlying NAFLD pathogenesis is poorly understood. In the current study, we identified a novel histone methylation-mediated mechanism involved in the development of hepatic steatosis.

Histone lysine demethylase KDM7A is a recently identified member of the PHD and JmjC domain-containing histone demethylase family [6]. It demethylates both H3K9me2 and H3K27me2, converting transcriptionally repressive histone marks to active histone marks. KDM7A was shown to regulate brain development, cell cycle, cell proliferation, and inflammation response [7,8,9]. However, its role in hepatic steatosis remains unclear. We used gain-of-function and loss-of-function studies in AML12 cells to determine whether KDM7A induces hepatic steatosis. Adenovirus-mediated KDM7A overexpression resulted in an increase in intracellular TG accumulation in AML12 cells, whereas shRNA-mediated KDM7A knockdown significantly decreased intracellular TG levels in AML12 cells. These findings suggest that KDM7A plays an important role in the development of hepatic steatosis. Then, we attempted to identify the target gene regulated by KDM7A by performing RNA-sequencing analysis. RNA-sequencing analysis revealed the upregulation in the expression of the genes associated with TG synthesis, including perilipin 5, monoacyl-transferase (MOAT), and DGAT1/2. Among these genes, DGATs catalyze the final step in the synthesis of TG through the esterification of diacylglycerol (DAG) with a fatty acid. Mammals have two isozymes of DGAT, DGAT1 and DGAT2 which have no sequence homology to each other. DGAT1 is expressed abundantly in the intestine and plays a role in the absorption of dietary lipids by reconstituting TG in a committed step. DGAT2 is mainly located in adipose tissue and liver and plays a central role in intracellular TG accumulation through esterification of endogenous fatty acid to DAG. Several reports have demonstrated that DGAT2 contributes to the development of hepatic steatosis [13,14,15,16]. DGAT2 antisense-treated mice and hepatic-specific DGAT2 knockout mice have lower TG accumulation and ameliorated hepatic steatosis [12,13,14]. Therefore, we believe that DGAT2 may be a potential downstream target gene of KDM7A necessary for the development of hepatic steatosis. In the present study, we observed that KDM7A overexpression increased DGAT2 expression and intracellular TG accumulation in AML12 cells. Enzymatic inhibition of DGAT2 using PF-06427878 significantly blocked the KDM7A-mediated-increase in TG accumulation, suggesting that DGAT2 is a downstream target gene of KDM7A. In contrast, KDM7A knockdown resulted in decreased DGAT2 expression and TG accumulation in AML12 cells and significantly reduced FFAs-induced intracellular TG accumulation. Together, these results indicate that KDM7A promotes DGAT2 expression, leading to increased intracellular TG accumulation, and consequently results in the development of hepatic steatosis.

We ascertained the functional role of KDM7A in the pathogenesis of hepatic steatosis in vivo. We found that mice injected with recombinant Ad-KDM7A showed elevated hepatic TG accumulation as revealed from the quantification of TG levels and ORO and H&E staining. Thus, KDM7A contributes to the development of hepatic steatosis. Furthermore, we confirmed KDM7A-DGAT2 signaling in the liver tissue of Ad-KDM7A-injected mice. Consistent with the results in AML12 cells, the expression of DGAT2 was higher in the liver of Ad-KDM7A-injected mice than in the liver of Ad-GFP-injected mice. Therefore, the in vivo data show that KDM7A is an epigenetic regulator involved in the development of hepatic steatosis through the upregulation of DGAT2 expression.

NASH develops from hepatic steatosis by a combination of TG accumulation, apoptosis, oxidative stress, and inflammation [1]. It is uncertain whether the KDM7A-induced hepatic steatosis shown in the present study favors the development of NASH and liver injuries. Hepatic steatosis is a driving force for NAFLD because increased intracellular TG accumulation induces inflammation, oxidative stress and insulin resistance by generating lipotoxic intermediates, which leads to NASH, fibrosis and cirrhosis [2]. On the other hand, hepatic TG synthesis from toxic saturated fatty acids prevents NAFLD development [4]. Recently, it was reported that overexpression of histone demethylase Phf2, which belongs to the KDM7 histone demethylase family, favored hepatic steatosis development, but protected the liver from inflammation and oxidative stress by desaturation of detrimental saturated fatty acid into DAG and TG, and fibrosis development [4].

Furthermore, the contradictory role of DGAT2 during hepatic steatosis progression to NASH has been reported [13,14,15]. Amin et al. showed that DGAT2 inhibition prevented the development of hepatic steatosis and steatohepatitis because it inhibited TG synthesis [13]. However, other studies reported opposite results [13,14]. Hepatic deletion of DGAT2 reduced hepatic steatosis but showed no increase in inflammation and fibrosis in mice [14]. Further, DGAT2 deficiency using antisense oligo improved hepatic steatosis but exacerbated liver damage and hepatic fibrosis [15]. Therefore, to clarify whether KDM7A-DGAT2 signaling-induced hepatic steatosis exacerbates or prevents the development of NASH and fibrosis, further studies should be warranted.

We determined the mechanism underlying KDM7A-mediated upregulation in DGAT2 expression. Di-methylation of H3K9 (H3K9me2) and H3K27 (H3K27me2) is associated with transcriptional repression of genes. To characterize the mechanism by which KDM7A promotes DGAT2 expression, we examined the changes in H3K9me2 and H3K27me2 levels on the DGAT2 promoter in KDM7A-overexpressing AML12 cells using ChIP-PCR. The recruitment of KDM7A to the promoter of DGAT2 was increased, whereas the enrichment of H3K9me2 and H3K27me2 was reduced in KDM7A-overexpressing AML12 cells. These results suggest that KDM7A erases repressive histone marks H3K9me2 and H3K27me2 on the promoter of DGAT2, resulting in the stimulation of DGAT2 expression.

Based on these findings, we propose an epigenetic model of KDM7A associated with the development of hepatic steatosis. Under the normal state, di-methylation of H3K9 (H3K9me2) and H3K27 (H3K27me2) on the promoter of DGAT2 represses DGAT2 expression and reduces hepatic TG accumulation. However, under hepatic steatosis states such as obesity, KDM7A is overexpressed and erases repressive H3K9me2 and H3K27me2 on the promoter of DGAT2, which leads to increased TG accumulation and results in hepatic steatosis (Figure 6). Accordingly, the present study provides a new insight into the potential role of epigenetic regulation of hepatic steatosis.

## 4. Materials and Methods

### 4.1. Reagents

Dulbecco’s modified Eagle’s medium/Ham’s Nutrient Mixture F-12 1:1(DMEM/F-12), penicillin–streptomycin, and fetal bovine serum were obtained from HyClone Laboratories Inc. (Logan, UT, USA). Oleic acid, palmitic acid and PF-06427878 were purchased from Sigma-Aldrich (St. Louis, MO, USA). The antibody against KDM7A was purchased from Aviva systems biology (San Diego, CA, USA). Antibodies against H3K9me2 and H3K27me2 were obtained from Millipore (Billerica, MA, USA). Antibodies against DGAT2 was obtained from Novus Biologicals (Littleton, CO, USA) and β-actin was supplied by Santa Cruz Biotechnology (Santa Cruz, CA, USA).

### 4.2. Cell Culture

HepG2 and AML12 cells were obtained from the American Type Culture Collection (Manassas, VA, USA) and cultured in DMEM/F12 1:1 supplemented with 10% heat-inactivated FBS, 1X Insulin-Transferrin-Selenium (GibcoTM, Paisley, UK), 40 ng/mL dexamethasone, 20 U/mL penicillin, and 20 μg/mL streptomycin. HepG2 cells and AML12 cells were treated with 1 mM a mixture of free fatty acids (FFAs) (oleic acid: palmitic acid, 2:1) for 24 h to induce hepatic steatosis cell model.

### 4.3. Infection of AML12 Cells with Recombinant Adenovirus

Adenovirus vectors encoding green fluorescent protein (GFP), KDM7A (Ad-GFP, Ad-KDM7A), and containing scramble short-hairpin RNA (shRNA) and KDM7A shRNA (Ad-GFP shRNA, Ad-KDM7A shRNA) were purchased from Vector Biolabs (Malvern, PA, USA). AML12 cells were infected with Ad-GFP, Ad-KDM7A, Ad-GFP shRNA or Ad-KDM7A shRNA at a multiplicity of infection (MOI) ranging from 1 to 100 PFU/cell. Thereafter, the cells were washed and incubated with fresh medium. At 48 h post-infection, cells were harvested and analyzed.

### 4.4. Total RNA Preparation and Real-Time Quantitative Polymerase Chain Reaction (qPCR)

Total RNA was extracted using TRIzol (Invitrogen, Carlsbad, CA, USA) according to the manufacturer’s instructions. cDNA was generated from 1 μg of total RNA using the GoScript Reverse Transcription System (Promega, Madison, WI, USA) according to the manufacturer’s protocol. Real-time qPCR was performed using a SYBR Green premixed Taq reaction mixture with gene-specific primers; KDM7A sense, 5′-GAAATCAATCAGAAGGCACAAAG-3′; KDM7A antisense, 5′-GATTCAGCGTGTACTTTCCATTC-3′, and DGAT2 sense, 5′-AACCTGCTGACCACCAGGAACTAT-3′; DGAT2 antisense, 5′-AGGGCCTTATGCCAGGAAACTTCT-3′.

### 4.5. Western Blot Analysis

Equal amounts of proteins (20 μg/lane) from cell lysates or liver tissues were resolved on 10% sodium dodecyl sulfate polyacrylamide gel and then transferred onto polyvinylidene difluoride membranes (Millipore, Billerica, MA, USA). The membranes were blocked in 5% nonfat skim milk and probed with primary antibodies against KDM7A, H3K9me2, H3K27me2, DGAT2, and β-actin. After washing with Tween 20/Tris-buffered saline, the membranes were incubated with a horseradish peroxidase-conjugated secondary antibody (1:1000) for 1 h at room temperature. The proteins were detected using an enhanced chemiluminescence (ECL) western blot detection kit (Amersham, Uppsala, Sweden).

### 4.6. Chromatin Immunoprecipitation (ChIP)-qPCR

Briefly, AML12 cells were fixed with 1% formaldehyde for 10 min at room temperature. The crosslinked chromatin was sonicated to shear into 400-bp fragments using a Bioruptor sonicator (Diagenode, Denville, NJ, USA). Samples were immunoprecipitated using 1–2 μg antibodies against KDM7A, H3K9me2, H3K27me2, or nonspecific IgG control in the presence of a secondary antibody conjugated to Dynabeads (Invitrogen, Carlsbad, CA, USA). Purified DNA was subjected to qPCR using the following primers: DGAT2 sense, 5-GTCCAGCTGTAACCTAACGA-3; DGAT2 antisense, 5-AAGCCTCTCTCTCAGACGTT-3. ChIP data were normalized as per control IgG or expressed as a percentage of input.

### 4.7. Animal Experiments

C57BL/6 mice (8 weeks of age) were purchased from Central Lab Animal Inc. (Seoul, Korea) and maintained on a regular chow diet (ND: 10 kcal% fat, SLACOM, #M01). The mice were injected with a total of 1 × 10^9^ PFU recombinant adenoviruses (Ad-GFP or Ad-KDM7A) via tail vein injection. After injection, adenovirus-injected mice were fed a ND for 4 weeks. Four weeks after injection, the mice were fasted for 6 h, and blood was harvested from the tail vein. The animals were then sacrificed, and liver tissues were collected for further analysis. The animal protocol used in this study was reviewed and approved by the Pusan National University’s Institutional Animal Care and Use Committee in accordance with established ethical and scientific care procedures (PNU-2020-2718).

### 4.8. Histological Analysis

Liver tissues isolated from the mice were dissected and fixed in 10% buffered formalin. Fixed tissues were embedded in paraffin and sectioned at a 5 μm thickness using a frozen microtome (HM560H, Microm Laboratory, Walldorf, Germany). The liver sections were stained with hematoxylin and eosin (H&E) and Oil Red O (ORO) and subjected to microscopic observation.

### 4.9. Hepatic TG Measurement

Hepatic lipids were extracted from cell extracts and liver tissues according to the following procedure. Briefly, liver tissues were homogenized in a chloroform-methanol solution (2:1 *v*/*v*). The homogenates were incubated for 1 h at room temperature and centrifuged at 3000 rpm for 10 min. The bottom layer (organic phase) was removed and dried overnight. After dissolving the sample in ethanol, hepatic TG level was determined using an AM 157S-K TG kit (Asan Pharmaceuticals Co., Ltd., Seoul, Korea) and normalized to the protein concentration.

### 4.10. Blood Biochemistry Analysis

After fasting for 6 h, the mice were sacrificed, and blood was collected. The samples were centrifuged at 1000× *g* for 15 min at 4 °C to obtain the serum and stored at −80 °C until analysis. The levels of TG, free fatty acids and total cholesterol in serum were determined using a DOGEN Kit (DoGenBio. Co. Ltd., Seoul, Korea).

### 4.11. Statistical Analysis

Data are expressed as the mean ± standard deviation (SD). Statistically significant differences were determined using a two-tailed Student’s *t*-test. For all statistical analyses, values of *p* < 0.05 were considered significant.

## 5. Conclusions

In conclusion, KDM7A plays a pivotal role in the development of hepatic steatosis by upregulating DGAT2 expression. The results of the present study provide a novel insight into the potential epigenetic regulation of hepatic steatosis.

## Figures and Tables

**Figure 1 ijms-22-11085-f001:**
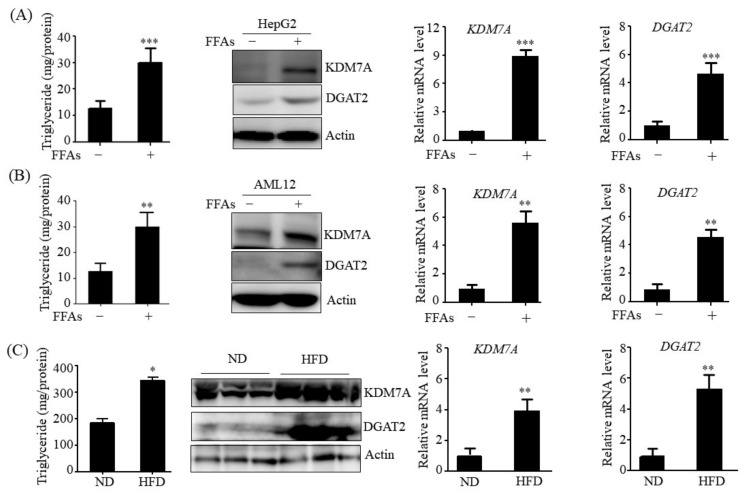
The expression of KDM7A and DGAT2 is increased in FFAs-treated HepG2 and AML12 cells and in the liver of HFD-fed obese mice. (**A**,**B**) HepG2 and AML-12 cells were incubated with 1 mM FFAs for 24 h. TG levels were measured by a TG assay kit. Protein levels of KDM7A and DGAT2 were measured by Western blotting. mRNA levels of KDM7A and DGAT2 were measured by qPCR. Data are presented as the mean ± SD from three independent experiments. ** *p* < 0.01, *** *p* < 0.001 vs. untreated control. (**C**) C57BL/6 mice were fed a normal diet (ND) or high fat diet (HFD) for 16 weeks. Hepatic TG levels were measured by a TG assay kit. Protein levels of KDM7A and DGAT2 were assayed by Western blotting. Representative Western blot analysis of KDM7A and DGAT2 is shown. mRNA level of KDM7A was measured by qPCR. Data are presented as the mean ± SD from 6 mice per group. * *p* < 0.05, ** *p* < 0.01 vs. ND mice.

**Figure 2 ijms-22-11085-f002:**
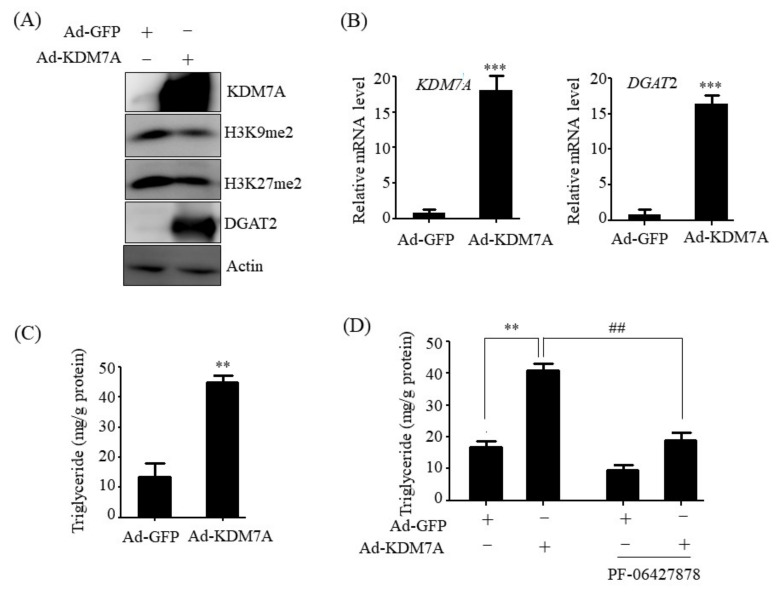
KDM7A overexpression increases DGAT2 expression and intracellular TG levels in AML12 cells. AML12 cells were infected with Ad-GFP or Ad-KDM7A. (**A**) Protein levels of KDM7A, H3K9me, H3K27me2 and DGAT2 were measured by Western blotting. (**B**) mRNA levels of KDM7A and DGAT2 were measured by qPCR. (**C**) Intracellular TG levels were measured using a TG assay kit. Data are presented as the mean ± SD from three independent experiments. ** *p* < 0.01, *** *p* < 0.001 vs. control (Ad-GFP). (**D**) AML12 cells were infected with Ad-GFP or Ad-KDM7A and then treated with PF-06427878 for an additional 24 h. Intracellular TG levels were measured using a TG assay kit. Data are presented as the mean ± SD from three independent experiments. ** *p* < 0.01 vs. control (Ad-GFP), ^##^
*p* < 0.01 vs. Ad-KDM7A without PF-06427878 treatment.

**Figure 3 ijms-22-11085-f003:**
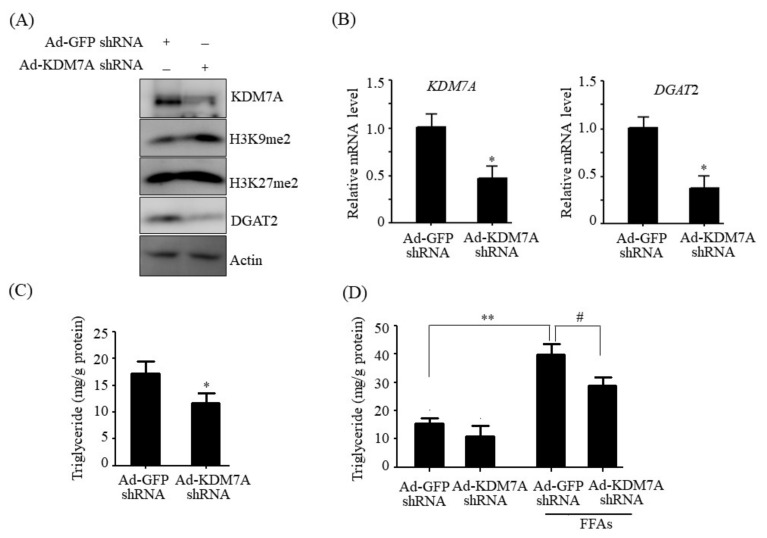
KDM7A knockdown decreases DGAT2 expression and TG level in AML12 cells. AML12 cells were infected with Ad-GFP shRNA or Ad-KDM7A shRNA. (**A**) Protein levels of KDM7A, H3K9me, H3K27me2 and DGAT2 were measured by Western blotting. (**B**) mRNA levels of KDM7A and DGAT2 were measured by qPCR. (**C**) Intracellular TG levels were measured using a TG assay kit. Data are presented as the mean ± SD from three independent experiments. * *p* < 0.05 vs. control (Ad-GFP shRNA). (**D**) AML12 cells were infected with Ad-GFP shRNA or Ad-KDM7A shRNA and then incubated with 1 mM FFAs for an additional 24 h. Intracellular TG levels were measured using a TG assay kit. Data are presented as the mean ± SD from three independent experiments. ** *p* < 0.01 vs. control (Ad-GFP shRNA). ^#^
*p* < 0.05 vs. FFAs-treated Ad-GFP shRNA.

**Figure 4 ijms-22-11085-f004:**
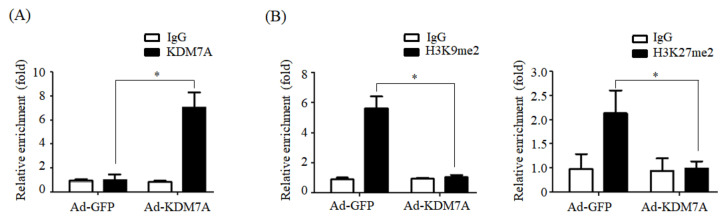
KDM7A overexpression reduces the enrichment of H3K9me2 and H3K27me2 on DGAT2 promoter. AML12 cells were infected with adenovirus Ad-GFP or Ad-KDM7A and ChIP-PCR was performed. (**A**) The recruitment of KDM7A to the DGAT2 promoter was analyzed by ChIP-qPCR. (**B**) The enrichment of H3K9me2 and H3K27me2 on the promoter of DGAT2 was analyzed by ChIP-qPCR. Data are presented as the mean ± SD from three independent experiments. * *p* < 0.05 vs. control (Ad-GFP).

**Figure 5 ijms-22-11085-f005:**
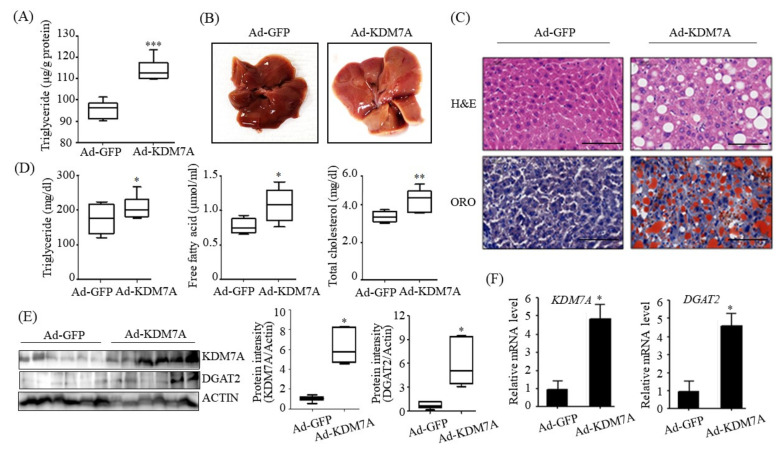
Adenovirus-mediated KDM7A overexpression induces hepatic steatosis in vivo. C57BL/6 mice (8 weeks old) were injected with adenovirus Ad-GFP or Ad-KDM7A. After injection, Ad-injected mice were fed a ND for 4 weeks. (**A**) Quantitation of hepatic TG. Hepatic TG levels were measured using a TG assay kit. (**B**) Representative photograph of liver. (**C**) Representative H&E and ORO staining images (scale bar = 50 μm). (**D**) Serum levels of TG, free fatty acid and total cholesterol. (**E**) Protein levels of KDM7A and DGAT2 were determined by Western blotting. Quantifications were performed using ImageJ from 6 mice. (**F**) mRNA levels of KDM7A and DGAT2 were measured by qPCR. Data are presented as the mean ± SD from 6 mice per group. * *p* < 0.05, ** *p* < 0.01, *** *p* < 0.001 vs. control mice (Ad-GFP).

**Figure 6 ijms-22-11085-f006:**
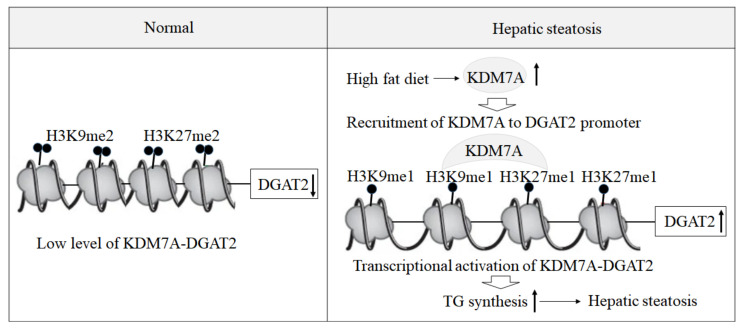
A proposed epigenetic model of KDM7A-DGAT2 in the development of hepatic steatosis. Up arrow means upregulation and down arrow means downregulation.

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
