# Peer review of "Histone Demethylase KDM7A Contributes to the Development of Hepatic Steatosis by Targeting Diacylglycerol Acyltransferase 2"

_ijms, 2021, doi:10.3390/ijms222011085_

Round 1
Reviewer 1 Report
In this manuscript, the authors showed how they had studied the functional role of KDM7A in the development of hepatic steatosis. They showed that in the hepatosteatosis cell and animal models, the expression levels of KDM7A and DGAT2 significantly increased. Then gain-of-function studies were performed and showed that overexpression of KDM7A resulted in increased triglyceride and DGAT2 and decreased H3K9me2 and H3K27me2, while KDM7A knockdown led to decreased triglyceride and DGAT2. ChIP-qPCR assay was performed to show that increased recruitment of KDM7A onto DGAT2 promoter caused reduced enrichment of H3K9me2 and H3K27me2. Furthermore, KDM7A was overexpressed in mice and it was observed that compared to the control group, the mice with overexpressed KDM7A developed increased steatosis as well as increased DGAT2 in the liver. Taken these results together, the authors proposed a model that increased KDM7A would decrease H3K9me2 and H3K27me2 to increase DGAT2 expression to increase triglyceride level and contribute to the development of hepatic steatosis.
The studies presented in this manuscript contribute to a better understanding of the mechanism by which KDM7A regulates hepatic steatosis, the simplest stage of the nonalcoholic fatty liver disease which is one of the most common chronic liver disease. Experiments were done in both cell and animal models. The data were straight forward and sufficient controls were included. The manuscript was well written and organized.
I only have one minor suggestion: it was described in the Materials and Methods section (line 363) that for Western blot sample loading, equal amounts of proteins were loaded. And the gel pictures show that Actin was used as the loading control. Different from expectation,it is noticeable that in Fig. 5E, the signals for Actin were different between the Ad-GFP control and the Ad-KDM7A treated group, and even quite variable among the 6 mice samples in the Ad-KDM7A treated group. Since this is not observed in other Western blot pictures which also assess the amounts of KDM7A, DGAT2 using the Actin control, some explanation or clarification is expected for this Fig. 5E.
Author Response
(1) It was described in the Materials and Methods section (line 363) that for Western blot sample loading, equal amounts of proteins were loaded. And the gel pictures show that Actin was used as the loading control. Different from expectation,it is noticeable that in Fig. 5E, the signals for Actin were different between the Ad-GFP control and the Ad-KDM7A treated group, and even quite variable among the 6 mice samples in the Ad-KDM7A treated group. Since this is not observed in other Western blot pictures which also assess the amounts of KDM7A, DGAT2 using the Actin control, some explanation or clarification is expected for this Fig. 5E.
Response;
- Thank you very much for comments. We extracted proteins from the liver tissue, then determined the concentration of proteins using Bradford assay and BCA assay, and loaded same amounts of proteins on the gel.
Despite multiple experiments, we observed the difference in Actin.
We believe that the overexpression of KDM7A could result in the accumulation of TG in the liver, which affected the liver lysates of Ad-KDM7A-injected mice and led to difference in Actin, unlike control mice (Ad-GFP mice).
Therefore, in order to improve the concern about difference in Actin as loading control, we quantified the band density of each protein on the gel using ImageJ, and represented the intensity ratio for KDM7A, DGAT2/Actin, which was shown on the right side of the western blotting in Fig.5E.
Reviewer 2 Report
The work "Histone demethylase KDM7A contributes to the development 2 of hepatic steatosis by targeting diacylglycerol acyltransferase " is exciting and novel since the authors chose a focused work with histone demethylase KDM7A.
The fundamental aspect of NAFLD or steatosis lies in ectopic lipid deposition in tissues where it is not stored. In essence, this is a problem of lipid metabolic diseases where TG are an immediate, measurable index. Although the present work is finely crafted to examine the role of an epigenetic regulator in modulating this lipid metabolic disease, the authors paid less attention to the core metabolic and inflammatory roles of histone demethylase KDM7A.
The functional assays of KDM7A are nicely executed and examined. The abstract is highlighted too much about the methodological data-driven statement without less about the physiological and pathological relevance of steatosis and free fatty acid metabolism.
The cellular model of oleate and palmitate mixture in this proportion does not represent steatosis in which saturated FFAs promote an acute harmful effect of fat overaccumulation in the liver.
Line 400- how much liver tissue was used for TG estimation
Line 396 cryosectioning is often used for oil red exp, while H&E staining can be done at room temp. Did the authors used similar conditions for these two distinct assays
Line 387 It is not clear why four weeks was considered to develop the symptom of steatosis
Line 343 The cellular model of oleate and palmitate mixture in this proportion does not represent steatosis in which saturated FFAs promote an acute harmful effect of fat overaccumulation in the liver. Instead, it means minor toxic and apoptotic effects, thus representing a cellular model of steatosis that mimics benign chronic steatosis (Chem Biol Interact . 2007 Jan 30;165(2):106-16. doi: 10.1016/j.cbi.2006.11.004. Epub 2006 Nov 23).
Unlike palmitic acid (SFA), which accumulates intercellularly, oleic acid (MUFA) oxidised partially. The rationale to choose the FFA proportion used in this study is not clear. The expression of DGAT2 should be measured in the presence of palmitic acid as a control of the FFA mixture.
Line 214 how the appearance of the liver revealed mice increased steatosis (Fig. 5B). Authors need to characterize the "appearance" with reference to physiological changes of liver pathology.
The target gene expression was silenced by 50% by shRNA (Fig.3B), which is also close (~ 40%) to the suppression of non-specific gene DGAT2. What is the efficiency of gene-specific knockdown?
Inadequate previous work is cited in the subject with reference to both steatosis and epigenetics regulation in combinations or alone. There is no work cited beyond 2020.
There are a few recent related works are included as a reference.
Biomedicines . 2021 Sep 18;9(9):1256. doi: 10.3390/biomedicines9091256
Shen X, Liang X, Ji X, You J, Zhuang X, Song Y, Yin H, Zhao M, Zhao L. CD36 and DGAT2 facilitate the lipid-lowering effect of chitooligosaccharides fatty acid intake and triglyceride synthesis signaling. Food Funct. 2021 Sep 20;12(18):8681-8693. doi: 10.1039/d1fo01472b. PMID: 34351342.
Gastaldelli A, Stefan N, Häring HU. Liver-targeting drugs and their effect on blood glucose and hepatic lipids. Diabetologia. 2021 Jul;64(7):1461-1479. doi: 10.1007/s00125-021-05442-2. Epub 2021 Apr 20. PMID: 33877366; PMCID: PMC8187191.
Loomba R, Morgan E, Watts L, Xia S, Hannan LA, Geary RS, Baker BF, Bhanot S.Novel antisense inhibition of diacylglycerol O-acyltransferase 2 for treatment of non-alcoholic fatty liver disease: a multicentre, double-blind, randomised,placebo-controlled phase 2 trial. Lancet Gastroenterol Hepatol. 2020Sep;5(9):829-838. doi: 10.1016/S2468-1253(20)30186-2. Epub 2020 Jun 15. PMID:32553151.
Amin NB, Carvajal-Gonzalez S, Purkal J, Zhu T, Crowley C, Perez S, Chidsey K,Kim AM, Goodwin B. Targeting diacylglycerol acyltransferase 2 for the treatmentof nonalcoholic steatohepatitis. Sci Transl Med. 2019 Nov 27;11(520):eaav9701.doi: 10.1126/scitranslmed.aav9701. PMID: 31776293.
Author Response
(1) The functional assays of KDM7A are nicely executed and examined. The abstract is highlighted too much about the methodological data-driven statement without less about the physiological and pathological relevance of steatosis and free fatty acid metabolism.
Response;
- Thank you very much for your valuable comments. According to your suggestion, we deleted methodological data-driven statements in the Abstract.
(2) The cellular model of oleate and palmitate mixture in this proportion does not represent steatosis in which saturated FFAs promote an acute harmful effect of fat overaccumulation in the liver.
Response;
- As far as our knowledge, numerous previous studies (ref. 1 and 2) have shown that liver cells were treated with a combination of oleate and palmitate for hepatic steatosis in vitro cell model, and HFD induced obese mice was used for hepatic steatosis in vivo model.
Thus, in our current study, 1 mM FFAs mixtures of oleate and palmitate (2:1 ratio) was used to make hepatic steatosis in vitro cell model, because 1 mM FFAs mixtures (oleate and palmitate, 2:1 ratio) gave less toxic and apoptotic effects than palmitate alone, and induced intracellular fat accumulation.
- Galangin Improved Non-Alcoholic Fatty Liver Disease in Mice by Promoting Autophagy.Zhang X, Deng Y, Xiang J, Liu H, Zhang J, Liao J, Chen K, Liu B, Liu J, Pu Y.Drug Des Devel Ther. 2020 Aug 19;14:3393-3405. doi: 10.2147/DDDT.S258187. eCollection 2020.
- Picroside II attenuates fatty acid accumulation in HepG2 cells via modulation of fatty acid uptake and synthesis.Dhami-Shah H, Vaidya R, Udipi S, Raghavan S, Abhijit S, Mohan V, Balasubramanyam M, Vaidya A. Clin Mol Hepatol. 2018 Mar;24(1):77-87. doi: 10.3350/cmh.2017.0039. Epub 2017 Dec 19.
(3) Line 400- how much liver tissue was used for TG estimation
Response;
- 100 mg of liver tissue was used for TG estimation
(4) Line 396 cryosectioning is often used for oil red exp, while H&E staining can be done at room temp. Did the authors used similar conditions for these two distinct assays
Response;
- For histopathology, liver tissue samples from mice were collected and were divided into two groups, one for H&E staining and another for ORO staining. They were fixed in 10% buffered formalin, and small samples were snap frozen in liquid nitrogen. The samples fixed in formalin were dehydrated by an ascending alcohol series ending in xylol and finally embedded in paraffin. The specimens were sectioned at a thickness of 5 μm using a frozen microtome and sections were stained with H&E or ORO.
(5) Line 387 It is not clear why four weeks was considered to develop the symptom of steatosis.
Response;
- Because the overexpression of KDM7A was evident in Ad-KDM7A-injected mice at 4 weeks after injection. Furthermore, biochemical analysis of blood revealed that serum TG level was also increased at 4 weeks after injection.
(6) Line 343 The cellular model of oleate and palmitate mixture in this proportion does not represent steatosis in which saturated FFAs promote an acute harmful effect of fat overaccumulation in the liver. Instead, it means minor toxic and apoptotic effects, thus representing a cellular model of steatosis that mimics benign chronic steatosis (Chem Biol Interact . 2007 Jan 30;165(2):106-16. doi: 10.1016/j.cbi.2006.11.004. Epub 2006 Nov 23).
Unlike palmitic acid (SFA), which accumulates intercellularly, oleic acid (MUFA) oxidised partially. The rationale to choose the FFA proportion used in this study is not clear.
The expression of DGAT2 should be measured in the presence of palmitic acid as a control of the FFA mixture.
Response;
- The FFAs mixtures we used in the current study was 1 mM FFAs mixtures of oleate and palmitate at 2:1 ratio. The previous study (Chem Biol Interact . 2007 Jan 30;165(2):106-16.) that you referred in the comment supports that this condition is associated with minor toxic and apoptotic effects. Thus, we used the same condition to represent a cellular model of steatosis that mimics benign chronic steatosis.
- In preliminary study, we already observed that treatment of HepG2 cells with palmitate also enhanced DGAT2 expression.
(7) Line 214 how the appearance of the liver revealed mice increased steatosis (Fig. 5B). Authors need to characterize the "appearance" with reference to physiological changes of liver pathology.
Response;
- As shown in Fig. 5B, the liver tissue of Ad-KDM7A-injected showed a light brown color, compared to healthy liver tissue of control mice (Ad-GFP mice) with dark brown color, which means high hepatic TG accumulation.
In addition, analysis of hepatic TG level and ORO staining revealed higher TG levels and more lipid droplets in Ad-KDM7A injected mice than those in control mice (Ad-GFP mice).
(8) The target gene expression was silenced by 50% by shRNA (Fig.3B), which is also close (~ 40%) to the suppression of non-specific gene DGAT2. What is the efficiency of gene-specific knockdown?
Response;
- The efficiency of shRNA against KDM7A was approximately 40%.
Since DGAT2 is a downstream target gene of KDM7A based on the results of our current study, the expression of DGAT2 was also decreased up to 40% by shRNA-mediated KDM7A silence.
(9) Inadequate previous work is cited in the subject with reference to both steatosis and epigenetics regulation in combinations or alone. There is no work cited beyond 2020. There are a few recent related works are included as a reference.
Response;
- According to your suggestion, we add three recent references (ref 12, ref 16, ref 17) below in Discussion.
Ref 12 ; Biomedicines . 2021 Sep 18;9(9):1256. doi: 10.3390/biomedicines9091256
Ref 16 ; Loomba R, Morgan E, Watts L, Xia S, Hannan LA, Geary RS, Baker BF, Bhanot S.Novel antisense inhibition of diacylglycerol O-acyltransferase 2 for treatment of non-alcoholic fatty liver disease: a multicentre, double-blind, randomised,placebo-controlled phase 2 trial. Lancet Gastroenterol Hepatol. 2020Sep;5(9):829-838. doi: 10.1016/S2468-1253(20)30186-2. Epub 2020 Jun 15. PMID:32553151.
Ref 17 ; Amin, N.B.; Carvajal-Gonzalez, S.; Purkal, J.; Zhu, T.; Crowley, C.; Perez, S.; Chidsey, K.; Kim, A.M.; Goodwin, B. Targeting diacylglycerol acyltransferase 2 for the treatmentof nonalcoholic steatohepatitis. Sci Transl Med. 2019,11.